# Deleting *Cellular Retinoic-Acid-Binding Protein-1* (*Crabp1*) Gene Causes Adult-Onset Primary Hypothyroidism in Mice

**Fatimah Najjar** [1] , **Jennifer Nhieu** [1] , **Chin-Wen Wei** [1] , **Liming Milbauer** [1] , **Lynn Burmeister** [2] , **Davis Seelig** [3] and **Li-Na Wei** [1,*]

1    Department of Pharmacology, University of Minnesota, Minneapolis, MN 55455, USA
2    Division of Diabetes, Endocrinology and Metabolism, Department of Medicine, University of Minnesota, Minneapolis, MN 55455, USA
3    Department of Veterinary, Comparative Pathology Shared Resource and College of Veterinary Medicine, Clinical Sciences, University of Minnesota, St. Paul, MN 55108, USA
*    Correspondence: weixx009@umn.edu; Tel.: +61-26-259-402

**Abstract:** Adult-onset primary hypothyroidism is commonly caused by iatrogenic or autoimmune mechanisms; whether other factors might also contribute to adult hypothyroidism is unclear. Cellular Retinoic-Acid-Binding Protein 1 (CRABP1) is a mediator for Non-canonical signalling of all-trans retinoic acid (atRA). CRABP1 Knockout (CKO) mice develop and reproduce normally but begin to exhibit primary hypothyroidism in adults (~3 months old) including increased body weight, decreased body temperature, reduced plasma levels of triiodothyronine and thyroxine, and elevated levels of thyroid-stimulating hormone. Histopathological and gene expression studies reveal significant thyroid gland morphological abnormalities and altered expression of genes involved in thyroid hormone synthesis, transport, and metabolism in the CKO thyroid gland at ~6 months old. These significantly affected genes in CKO mice are also found to be genetically altered in human patients with hypothyroidism which could result in a loss of function, supporting the clinical relevance of CKO mice in humans with hypothyroidism. This study identifies, for the first time, an important role for CRABP1 in maintaining the health of the thyroid gland in adults and reports that CKO mice may provide an experimental animal model for studying the mechanisms underlying the development of adult hypothyroidism in humans.

**Keywords:** hypothyroidism; thyroid hormones; thyroid gland; CRABP1; retinoic acid; vitamin A





## 1. Introduction

Thyroid hormone is critical for normal development as well as growth and the regulation of metabolic rate and thermogenesis [1]. The hypothalamus–pituitary–thyroid (HPT) axis regulates systemic thyroid hormone production via a negative feedback mechanism [2]. Thyroid hormone synthesis involves thyroid stimulating hormone (TSH) receptors, multiple transporters (NIS, pendrin, and MCT-8), and enzymes such as thyroid peroxidase (TPO) and dual oxidases (DUOX) working in a coordinated and regulated fashion to maintain thyroid homeostasis [3]. Thyroid-state-dependent regulation of peripheral thyroid hormone metabolism occurs to maintain triiodothyronine (T3) homeostasis. Deiodinases (DIOs) -1 and -2 enzymes convert thyroxine to the active form, T3 [4,5], and DIO-3 converts T4 to the inactive form, reverse triiodothyronine (rT3) [6].

Hypothyroidism is due to thyroid hormone deficiency. The prevalence of hypothyroidism in iodine-sufficient countries ranges from 1 to 7% in the general population, with the highest rates in the elderly [7]. Hypothyroidism may affect multiple organ systems, including goitres, weight gain, fatigue, cold intolerance, and other nonspecific symptoms. Primary hypothyroidism, the most common form of hypothyroidism, is caused by thyroid gland dysfunction, leading to thyroid hormone deficiency. The diagnosis is made with measurements of high thyroid-stimulating hormone (TSH) and low free thyroxine (T4)[8].

Primary hypothyroidism with onset in adult patients is commonly due to iatrogenic or autoimmune mechanisms. Genetic factors leading to non-autoimmune hypothyroidism have mainly been identified for congenital, or childhood-onset, hypothyroidism, including single gene mutations leading to thyroid dysgenesis or dyshormonogensis [9–11]. Mutations involving thyroid receptors, thyroid hormone-metabolizing enzymes, or thyroid transporters have been associated with specific alterations in plasma thyroid hormone levels and tissue-specific manifestations of thyroid deficiency [12–14].

Cellular retinoic-acid-binding protein-1 (CRABP1) is a high-affinity, cytosolic binding protein for *all-trans* retinoic acid (atRA), the principal active metabolite of vitamin A [15,16]. Molecular studies have revealed that CRABP1 plays major roles in atRA signalling in the cytoplasm, primarily through its ability to modulate important cytosolic enzymes involved in cell growth, such as the extracellular-signal-regulated kinase (ERK)–mitogen-activated protein kinase kinase (MEK)–mitogen-activated protein kinase (MAPK) (ERK-MEK-MAPK) pathway that responds to growth hormone stimulation, and in cellular function such as calcium/calmodulin-dependent protein kinase II (CaMKII) that is important for mediating and handling calcium signalling [17,18]. These CRABP1-mediated effects of atRA are generally referred to as "non-canonical" activities of atRA because their cytosolic localization and function are drastically distinct from the well-known genomic activities of atRA mediated by nuclear R.A. receptors (RAR) [16].

CRABP1 is highly expressed in the adult thyroid gland [19–21]. Our preliminary observations suggested an adult-onset hypothyroid phenotype in *Crabp1* gene knockout (CKO) mice. We, thus, designed studies to evaluate the specific defects in male CKO mice. This study reports that male CKO mice develop adult-onset primary hypothyroidism associated with thyrocyte defects consistent with a deficiency in thyroid hormone synthesis.

## 2. Materials and Methods

**Animals:** Wild-type C57BL/6 male mice were obtained from the Jackson Laboratory and housed in the University of Minnesota animal facilities. CKO male mice were maintained as previously described [17,22,23]. *Crabp1* gene was targeted by the hit and run technique; briefly, a 5 nt insertion between the second and third positions of the alanine codon of amino acid position 5 of CRABP1 exon 1 was made to create one Not I restriction site to ablate CRABP1 expression [17,24]. All experimental procedures were conducted according to NIH guidelines, and the protocols were approved by the University of Minnesota Institutional Animal Care and Use Committee. The animals were housed in a temperature- ($22 \pm 2$ °C) and light-controlled (12 h light/dark cycle; lights on at 6 a.m.) animal facility and had free access to food and water. Rectal body temperature was measured using a probe (RET-3) and digi-sense-temp thermometer (no.FF-91428-03, Cole-Parmer, Vernon Hills, IL, USA). Body weight and temperature were measured on three consecutive days in one week.

**Tissue collection:** Blood samples were collected from the submandibular vein, placed into an EDTA Minicollect tube (no.450480, GBO, Kremsmunster, Austria), and centrifuged at $10,000 \times g$ for 15 min at 4 °C. After $CO_2$ euthanasia, thyroid glands, and surrounding trachea were harvested. The major and minor diameters for each lobe of the thyroid gland were measured (a total of two lobes). The formula ($A = \pi \ast a \ast b$) was used to calculate the area of each lobe, where (a) represents the major radius and (b) represents the minor radius, and the sum of the two lobes was divided by the body weight in order to normalize the thyroid gland [25]. Afterward, the lobes of the thyroid gland were isolated and snap-frozen in liquid nitrogen and kept at −80 °C for protein and RNA studies. The resulting plasma and snap-frozen tissues were kept at −80 °C until further analyses.

**Hormonal assay:** Plasma samples were thawed and centrifuged at $10,000 \times g$ for 10 min at 4 °C. For each assay, 10 µL of plasma was utilized and the samples were analysed in duplicate. The hormonal assay was performed according to the following ELISA protocols: TSH ELISA kit (no.MBS162380, San Diego, CA, USA), total thyroxine ELISA kit (no.MBS580037, San Diego, CA, USA), total triiodothyronine (t3) ELISA kit

(no.GWB-74C5DC, San Diego, CA, USA), and reverse triiodothyronine (rT3) ELISA kit (no.MBS452357, San Diego, CA, USA).

**Hematoxylin and Eosin (H&E) staining:** Harvested thyroid glands were fixed in 10% formalin overnight and dehydrated in ethanol for paraffin block preparation. Tissue sections were cut at 5 μm and stained with haematoxylin and eosin stain for histopathological evaluation by a board-certified veterinary pathologist.

**Quantitative real-time PCR:** Frozen thyroid gland samples were homogenized in trizol (ambion) to extract mRNA, and 2 μg of total mRNA was reverse-transcribed to cDNA with reverse transcript kits (no. 4368819, Applied Biosystems by Thermo Fisher Scientific). A real-time PCR reaction was performed in triplicate for each sample. The values were normalized to GAPDH. All primers are described in Table 1.

**Table 1.** Primers used for quantitative real-time PCR.

| Gene Name | Primer Orientation | Primer Sequences | NCBI Reference |
|---|---|---|---|
| *Tshr* | Forward | GCTGTCGTTGAGTTTCCTCCAC | NM_011648.1 |
| | Reverse | CTGCTCTCATTACACATCAAAGAC | |
| *Thr-α* | Forward | CCTGGACAAAGACGAGCAGTGT | NM_178060.4 |
| | Reverse | CTGGATTGTGCGGCGAAAGAAG | |
| *Thr-β* | Forward | ACCACTATCGCTGCATCACCTG | NM_009380.3 |
| | Reverse | ACTGGTTGCGGGTGACTTTGTC | |
| *Dio1* | Forward | GTAGGCAAGGTGCTAATGACGC | NM_007860.4 |
| | Reverse | ACTGGATGCTGAAGAAGGTGGG | |
| *Dio2* | Forward | GGTGGTCAACTTTGGTTCAGCC | NM_010050.4 |
| | Reverse | AAGTCAGCCACCGAGGAGAACT | |
| *Dio3* | Forward | TGCGTATCAGACGACAACCGTC | NM_172119.2 |
| | Reverse | TGGAAGCCATCAGGTCGGACAA | |
| *TPO* | Forward | GAGAGGCTCTTCGTGCTGTCTA | NM_009417.1 |
| | Reverse | AGGCGTGACAAGCCACAGAACT | |
| *Duox1* | Forward | AGGCGTGACAAGCCACAGAACT | NM_001099297.1 |
| | Reverse | AGGCGTGACAAGCCACAGAACT | |
| *Duox2* | Forward | GAGAAAGGCTGTGACCAAGCAG | NM_177610.2 |
| | Reverse | TCACGCACTTGCTGGGATGAGT | |
| *Tg* | Forward | TTGTAGCCTGGAGAGTCAGCAC | NM_009375.2 |
| | Reverse | CACTGCACATCTTTCCTGGTGG | |
| *Slc5a5 (Nis)* | Forward | CATGCCATTGCTCGTGTTGGAC | NM_053248.2 |
| | Reverse | GCCATAGCGTTGATACTGGTGG | |
| *Slc16a2* | Forward | GTGTATTCCGCCAGCGCACTTA | NM_009197 |
| | Reverse | AAGAGCACCCAGGTCTCCTTGA | |
| *Slc16a10* | Forward | GGAGACAACCTATGCAGTGTGG | NM_001114332.1 |
| | Reverse | GCCAATGCACATGAAGAGCACC | |

**Statistical analysis:** To assess the effect of deleting *Crabp1* in CKO samples, a *t*-test was performed; to determine the progression of the disease, repeated measure ANOVA was performed, followed by a Bonferroni post hoc test. *p* values of $p < 0.005$ were considered statistically significant. All statistics were performed using the SPSS software.

### 3. Results

*3.1. Male CKO Mouse Phenotype Suggests Adult-Onset Hypothyroidism*

The CKO mice reached adulthood normally, without differences in growth rate or fertility compared to the W.T. mice. However, as shown in Figure 1A, the male CKO mouse body temperature was lower than W.T. at the adult age of 3 months, and the decrease persisted until the last time-point monitored, 12 months old. Likewise, as shown in Figure 1B, the CKO weight was higher than W.T. between 3 and 9 months of age, but not at 12 months. The CKO thyroid area (normalized to body weight) was significantly smaller than W.T., starting at approximately 6 months of age (Figure 1C). However, in older mice (12 months old), there was no difference.

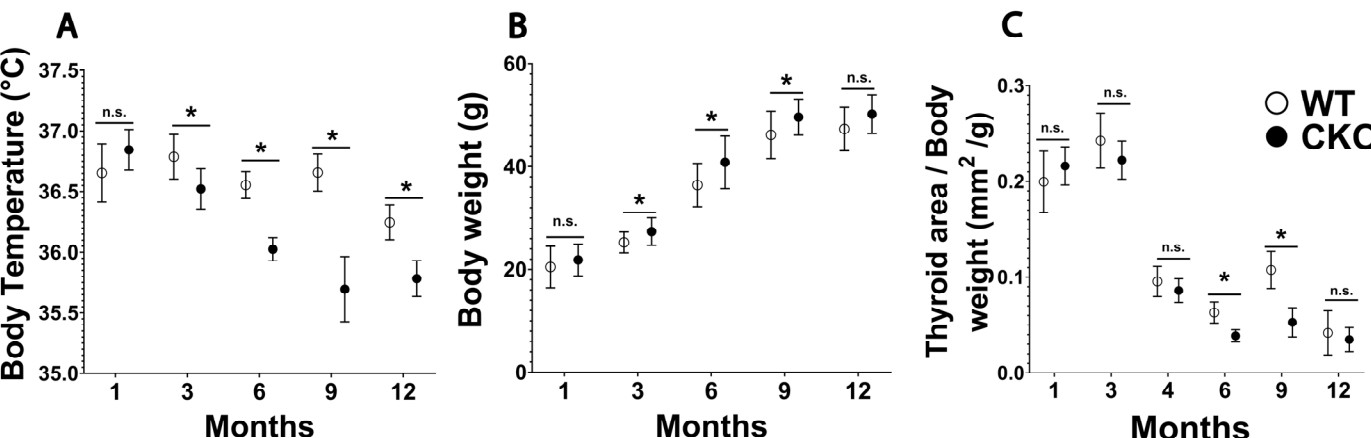

**Figure 1.** Physiological parameters of W.T. and CKO male mice at different ages. (**A**) Significant changes in CKO body temperature compared to W.T. at 3 months old [t (12) = 2.62, *p* = 0.022], 6 months old [t (48) = 3.51, *p* < 0.005], 9 months old [t (28) = 2.82, *p* < 0.005], and 12 months old [t (28), *p* = 0.032]. (**B**) Significant changes in CKO body weight compared to W.T. at 3 months old [t (10) = 2.23, *p* = 0.049], 6 months old [t (48) = 3.2, *p* < 0.005], and 9 months old [t (17) = 2.17, *p* = 0.042]. (**C**) Significant changes in CKO normalized thyroid area compared to W.T. at 6 months old [t (8) = 2.41, *p* = 0.042] and 9 months old [t (11) = 2.26, *p* = 0.045]. * *p* < 0.05 considered significant, CKO vs. W.T. male mice. n.s. (nonsignificant). Values are means ± S.E.; *n* = 7–9 mice/group. *t*-test was performed to compare CKO and W.T. male mice at 1, 3, 6, 9, and 12 months old. Repeated measures ANOVA as performed to monitor changes in body temperature, body weight, and the normalized thyroid area at different ages for each group.

### 3.2. Thyroid Hormone Status of CKO Mouse Suggesting Primary Hypothyroidism in Adults

Plasma thyroid hormone levels were measured between the 6- and 12-month time-points. T3 (Figure 2A) and T4 (Figure 2B) levels were significantly lower in the CKO mice at 6 and 9 months. The reverse T3 (rT3) level was dramatically and significantly elevated in the CKO mice between 6 and 12 months of age (Figure 2C). The male CKO mice had significantly elevated TSH levels starting at 5 months old, which persisted until the last time-point monitored, 12 months old (Figure 2D). Together, these data support the hypothesis that CKO mice develop adult-onset primary hypothyroidism, and they also have altered thyroid hormone metabolism.

### 3.3. Histopathology of CKO Thyroid Gland

To further evaluate the reduced thyroid area in the adult CKO mice, we assessed the histopathological changes in the CKO thyroid glands.

At an age of 1 month, the thyroid glands of the CKO and W.T. male mice appeared similar (Figure 3A,B). The thyroid glands contained well-differentiated colloid-filled folli-cles lined by morphologically unremarkable epithelium (thyrocytes). Small follicles were in the central zone, and larger follicles were in the peripheral areas. However, at 6 and 12 months old, the CKO thyroid glands appeared different from the comparable-age W.T. thyroid glands. The CKO thyroid gland had smaller follicles lined by larger epithelial cells (Figure 3D,F). The follicles were unevenly distributed throughout the peripheral and central zones. Both 6- and 9-month CKO thyroid glands had thyrocytes appearing like non-forming follicles.

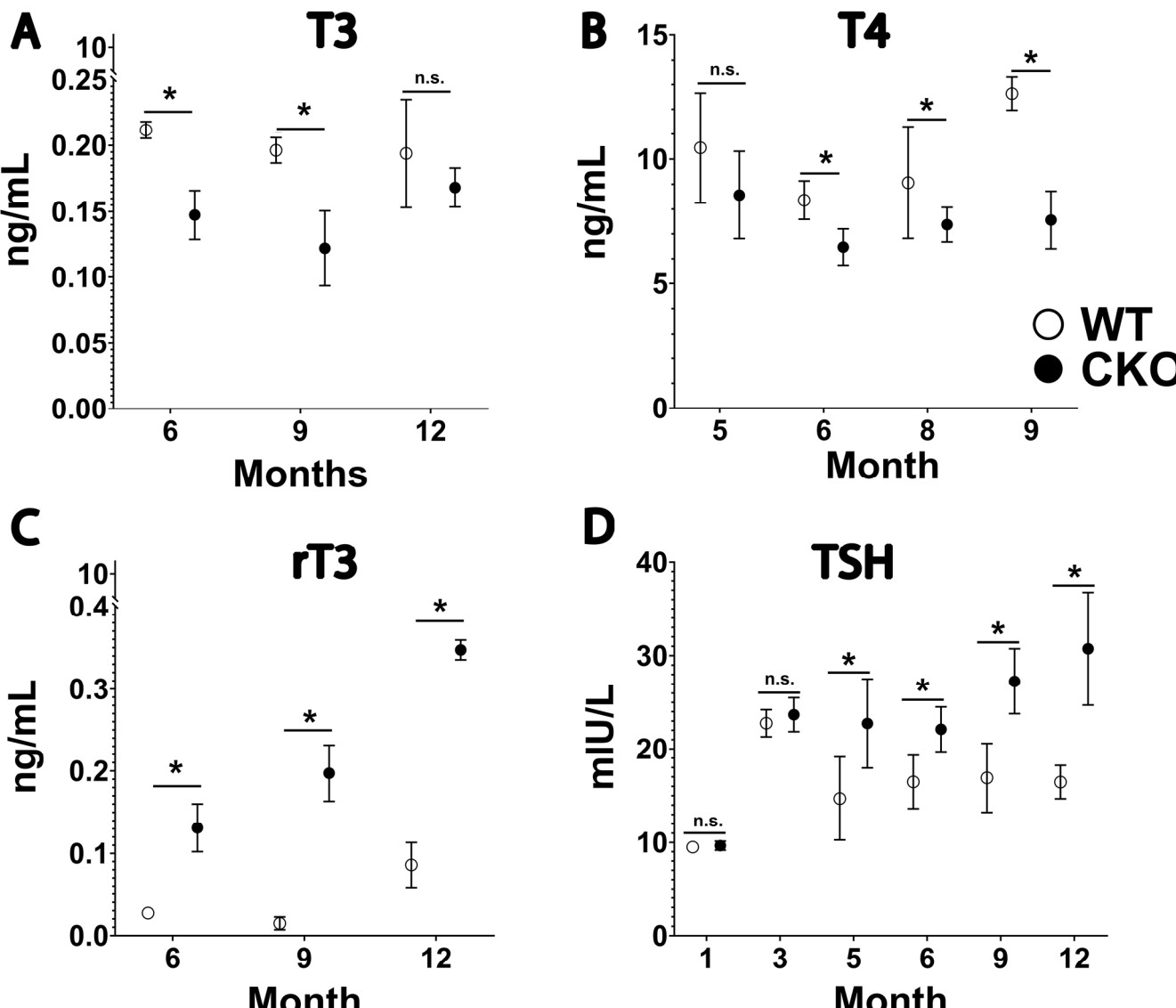

**Figure 2.** Plasma thyroid hormones and thyroid-stimulating hormone levels of CKO and W.T. male mice at different ages. (**A**) Significant changes in CKO T3 levels compared to W.T. at 6 months old [t (5) = 3.61, *p* = 0.015] and 9 months old [t (6) = 3.04, *p* = 0.023]. (**B**) Significant changes in CKO T4 levels at 6 months old [t (4) = 3.07, *p* = 0.037], 8 months old [t (4) = 4.54, *p* = 0.010], and 9 months old [t (4) = 6.56, *p* = 0.003]. (**C**) Significant changes in CKO rT3 levels compared to W.T. at 6 months old [t (6) = 6.54, *p* = 0.014], 9 months old [t (6) = 2.69, *p* = 0.036], and 12 months old [t (6) = 5.74, *p* < 0.005. (**D**) Significant changes in CKO TSH levels compared to W.T. at 5 months old [t (3) = 2.9, *p* <0.005], 6 months old [t (11) = 3.45, *p* = 0.005], 9 months old [t (5) = 3.95, *p* =0.011], and 12 months old [t (10) = 2.12, *p* < 0.005]. * *p* < 0.05 considered significant, CKO vs. W.T. male mice. n.s. (nonsignificant). Values are means ± S.E.; *n* = 3–5 mice/group. *t*-test was performed to compare CKO vs. W.T. male mice at 5, 6, 9, and 12 months. Repeated measures ANOVA was performed to monitor changes in TSH, T4, T3, and rT3 levels for each group.

In contrast, the 6-month-old W.T. thyroid glands were populated by larger and more organized follicles that were lined by cuboidal-to-columnar epithelial cells. (Figure 3C). In the 12-month-old W.T. mice, the thyroid glands typically had larger follicles and more flattened thyrocytes as compared to the 6-month-old W.T. glands (Figure 3E), indicating inactive thyrocytes. Like the 6-month-old CKO mice, the 12-month-old CKO mice also had smaller (compared to W.T.) follicles (Figure 3F).

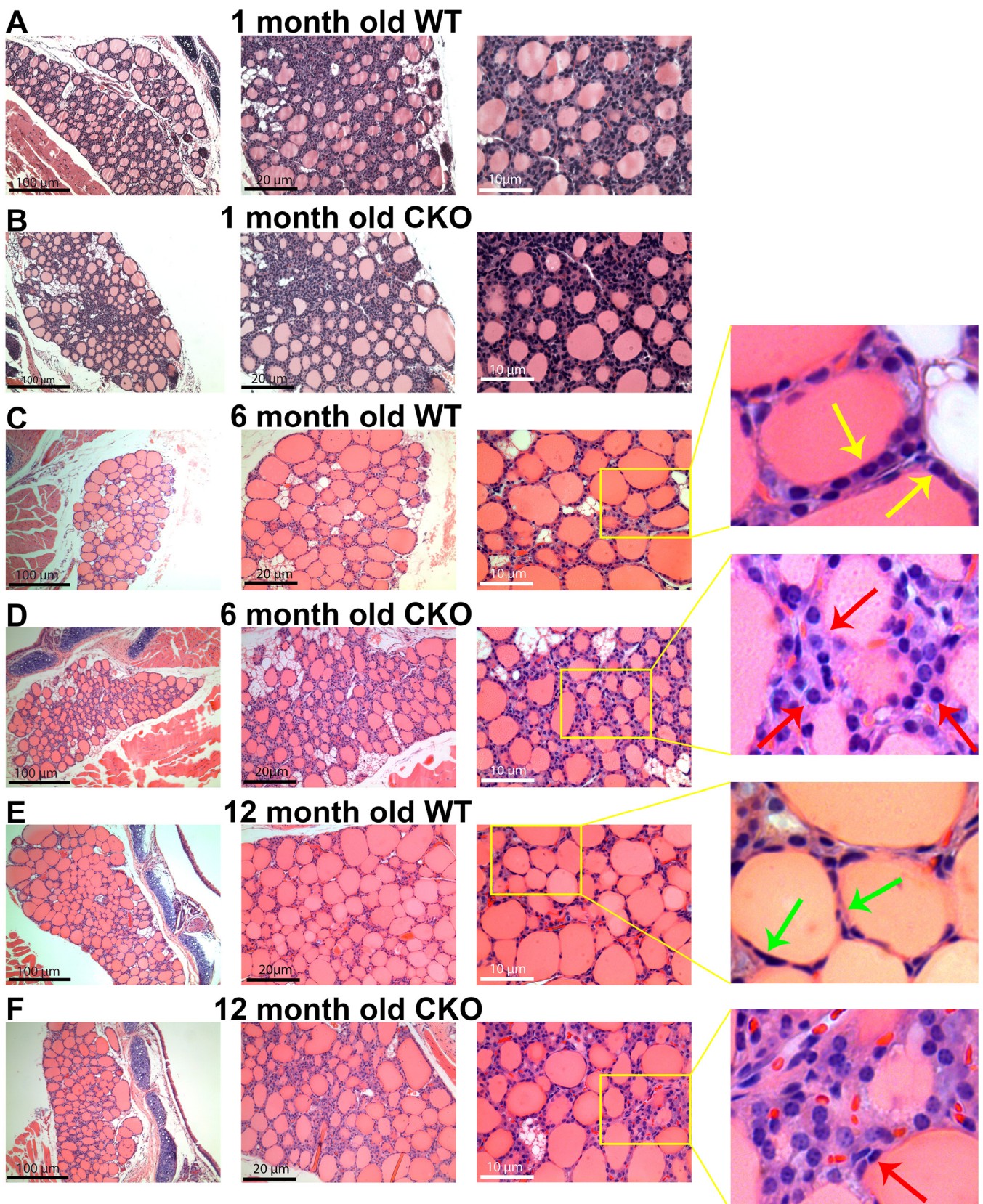

**Figure 3.** Hematoxylin and eosin (H&E) -stained thyroid gland sections from CKO and W.T. male mice. (**A,B**) 1-month-old W.T. and CKO mice showed no difference in the histology of the thyroid gland. (**C**) At 6 months old, W.T. mice had small follicles in the central zone and intermediate-to-large-sized follicles in the peripheral zone. Furthermore, the thyrocytes exhibited a columnar-to-cuboidal

shape as pointed out by the yellow arrows in the enlarged image on the right, indicating that thyrocytes were active. (**D**) At 6 months old, CKO thyroid gland had smaller follicles which were irregularly distributed; furthermore, their thyrocytes appeared to be larger than W.T. thyrocytes (pointed out by red arrows in the enlarged image on the right). (**E**) In the 12-month-old W.T. thyroid gland, follicular size was larger than that of 6-month-old, and thyrocytes were flattened (pointed out by green arrows in the enlarged image on the right), indicating inactive thyrocytes. (**F**) The 12-month-old CKO thyroid gland had a similar feature as the 6-month-old CKO thyroid gland. Red arrow points to thyrocytes that are nonforming follicles in the enlarged image on the right.

### 3.4. Altered Gene Expression in CKO Thyroid Gland Related to Hypothyroidism

Potential defects that might explain the pathogenic mechanism(s) of adult-onset hypothyroidism in CKO mice were further explored by measuring the expression pattern of the genes involved in thyroid hormone synthesis, metabolism, and action. The studies were performed at the age of 8 months old, corresponding to the time of the most dramatic hormone alterations (between 6 and 9 months old). TSH receptor (TSH-R) expression was significantly down-regulated in the CKO thyroid gland, with no significant difference in the expression of thyroid receptors (TR) α or β (Figure 4A). The expression of thyroidal DIO-1 and -2 was significantly lower in the CKO mice, but no significant difference was detected for DIO-3 (Figure 4B). The thyroid hormone synthetic enzyme thyroid peroxidase (TPO) was down-regulated, dual oxidase 1 (DUOX-1) was up-regulated, DUOX-2 was not significantly altered, and there was a tendency towards an increase (although no statistical difference) in the thyroglobulin (T.G.) level in the CKO thyroid gland (Figure 4C). Lastly, we monitored transporters because of their roles in thyroid hormone homeostasis. The expression of both sodium iodide transporter (NIS) and thyroid hormone transporter, monocarboxylate transporter-8 (MCT-8), was down-regulated significantly in the CKO thyroid gland, but MCT-10 appeared unaffected (Figure 4D). Table 2 summarizes the changes in hormones, physiological parameters, and gene expression in adult male CKO mice aged 8–9 months old.

**Table 2.** Summary of changes in hormonal and physiological parameters and gene expression, comparing CKO and W.T. mice aged 8–9 months old.

| Variables | W.T. Male | CKO Male | |
| --- | --- | --- | --- |
| **Hormonal changes** | | | |
| TSH (mIU/L) | $16.9 \pm 0.74$ | $27.3 \pm 0.87$ | Increased |
| T4 (ng/mL) | $12.6 \pm 0.22$ | $7.6 \pm 0.39$ | Decreased |
| T3 (ng/mL) | $0.19 \pm 0.004$ | $0.12 \pm 0.01$ | Decreased |
| rT3 (ng/mL) | $0.014 \pm 0.004$ | $0.20 \pm 0.02$ | Increased |
| **Physiological parameter changes** | | | |
| Body temperature (°C) | $36.7 \pm 0.05$ | $35.9 \pm 0.09$ | Decreased |
| Body weight (g) | $44 \pm 0.54$ | $47 \pm 0.34$ | Increased |
| Normalized thyroid gland area ($mm^2/g$) | $0.06 \pm 0.005$ | $0.03 \pm 0.004$ | Decreased |
| **Changes in gene expression in the thyroid gland** | | | |
| TSH-R | $0.007 \pm 0.001$ | $0.003 \pm 0.001$ | Decreased |
| DIO-1 | $0.03 \pm 0.005$ | $0.01 \pm 0.001$ | Decreased |
| DIO-2 | $0.004 \pm 0.001$ | $0.001 \pm 0.001$ | Decreased |
| TPO | $0.09 \pm 0.02$ | $0.01 \pm 0.008$ | Decreased |
| DUOX-1 | $0.002 \pm 0.001$ | $0.004 \pm 0.001$ | Increased |
| NIS | $0.008 \pm 0.002$ | $0.003 \pm 0.001$ | Decreased |
| MCT-8 | $0.007 \pm 0.001$ | $0.004 \pm 0.001$ | Decreased |

Values are means $\pm$ S.E.

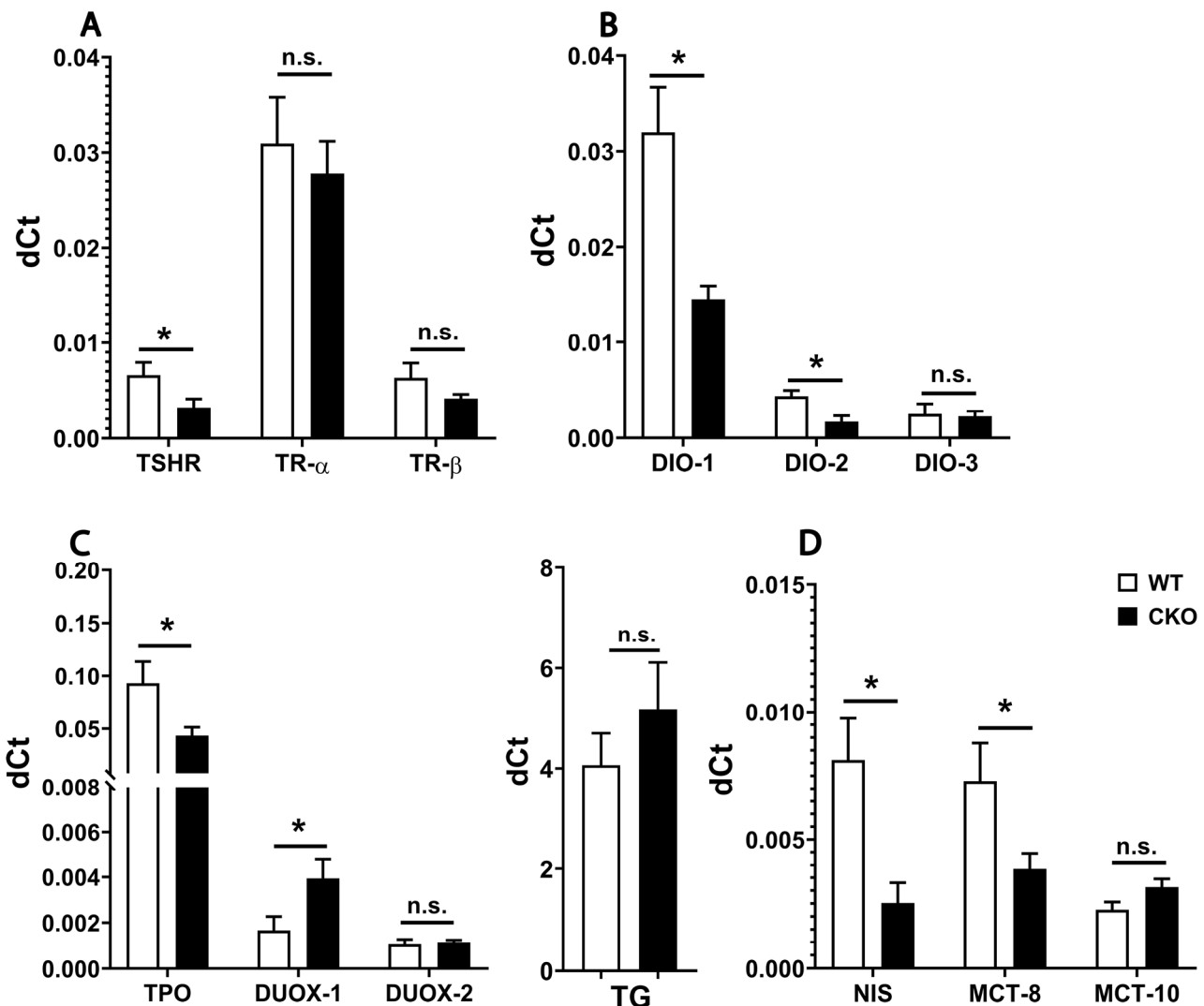

**Figure 4.** qPCR comparing the expression of specific genes in the thyroid gland between W.T. and CKO mice (8 months old). (**A**) The expression of various receptors. TSH-R showed significant reduction in CKO. [t (11) = 2.36, *p* = 0.035]. (**B**) The expression of DIO enzymes. DIO-1 showed significant reduction in CKO. [t (7) = 3.17, *p* = 0.016]. DIO-2 showed significant reduction in CKO. [t (8) = 2.95, *p* = 0.018]. (**C**) The expression of genes coding for enzymes involved in thyroid hormone synthesis. The expression of TPO was significantly reduced in CKO. [t (13) = 2.40, *p* = 0.032]. The expression of DUOX-1 was significantly reduced in CKO. [t (12) = 2.61, *p* = 0.023]. (**D**) The expression of iodide and thyroid hormone transporters. The expression of NIS was significantly reduced in CKO. [t (13) = 4.09, *p* < 0.005]. The expression of MCT-8 was significantly reduced in CKO. [t (6) = 3.03, *p* = 0.023, * *p* < 0.05 vs. W.T. male mice]. n.s.: nonsignificant. Values are means ± S.E.; *n* = 4–6 mice/group. *t*-test was performed to compare CKO and W.T. male mice.

### 3.5. Association between CKO's Molecular Changes and Human Genetic Defects in Hypothyroidism

The data from CKO mice shown above revealed significant changes in the expression of genes involved in thyroid hormone homeostasis in CKO mice, which could contribute to CKO hypothyroidism. The list includes TSHR, TPO, DUOX-1, NIS, and MCT-8 (Figure 4). To explore the potential clinical relevance of these molecular changes in CKO mice, we performed a literature review of these affected genes in human hypothyroidism. Table 3 summarizes the list of reported genetic alterations, ranging from single base-pair changes to complete exon deletions that ultimately resulted in a loss of function of the

genes, associated with human hypothyroidism. The disruption of protein function was validated through biochemical approaches or predicted as detrimental using in silico protein modelling [26–37]. Common to both human primary or tissue-specific hypothyroidism and CKO mice were genetic alterations in TSHR, TPO, NIS, DUOX-1, and MCT-8. Overall, the molecular changes detected in CKO mice were consistent with genetic alterations seen in humans where a loss of function of these proteins was highly implicated in, and/or directly causative of, hypothyroidism or altered metabolism. These data, together, suggest that CKO mice may be an appropriate animal model for understanding adult-onset hypothyroidism in humans.

**Table 3.** Genetic changes in human hypothyroidism consistent with altered gene expression in CKO mice.

| Gene Name | Genetic Alterations | Consequence | Reference |
|---|---|---|---|
| *TSHR* | <ul><li>Indel mutation in exon 1.</li><li>Missense mutations resulting in protein residue changes: R450H, G132R, A204V, D403N, or P556R.</li><li>Exon 2 deletion.</li></ul> | Loss of Function | [26–28] |
| *TPO* | <ul><li>2422delT, deletion mutation, resulting in an early stop codon at exon 14.</li><li>E378K missense mutation.</li><li>10 bp deletion at the intron 15–exon 16 boundary.</li><li>Missense mutations resulting in protein changes: S131P, N425S, and C838S.</li><li>Insertion GGCC bp insertion in exon 8, resulting in an early stop codon at exon 9.</li><li>20 bp duplication in exon 2, resulting in an early stop codon at exon 3.</li></ul> | Loss of Function | [29–33] |
| *NIS* | <ul><li>Missense mutation resulting in protein residue change T354P, G395R, G93R, G543E, or D267E.</li><li>67 bp deletion resulting in an early stop codon in exon 13.</li></ul> | Loss of Function | [34,35] |
| *MCT-8* | <ul><li>Single bp duplication in exon 5 resulting in frameshift mutation I539fs.</li></ul> | Loss of Function | [36] |
| *DUOX-1* | <ul><li>Missense mutation resulting protein residue change R56W.</li></ul> | Loss of Function | [37] |

Abbreviations: "SNP"—single nucleotide polymorphism, "bp", "fs"—frameshift.

Hypothyroidism or altered thyroid hormone metabolism has been associated with a wide spectrum of functions and potential disorders including neurodegenerative disease, cardiovascular diseases, diabetes mellitus type 2, infertility, etc. [1,38–41], which would suggest a potential crosstalk between thyroid hormone homeostasis and other organ systems. Table 4 [38,39,42–45] summarizes reported non-thyroid-related (extrathyroidal) hormone abnormalities detected in patients with hypothyroidism. Of particular interest are the association between hypothyroidism and alterations in RBP4 (vitamin A metabolism), hyper-beta-carotenemia (provitamin A), and adipokines (also altered in CKO mice) [23]. The presence of these extrathyroidal abnormalities in patients with hypothyroidism would strongly suggest a crosstalk between thyroid hormone homeostasis and other hormones, especially vitamin A, or R.A. signalling.

**Table 4.** Intersection of hypothyroidism, vitamin A signalling, and metabolic components.

| Hormone | Observation in Hypothyroidism | Reference |
|---|---|---|
| Beta-carotene | • Hyper-beta-carotenemia. | [42] |
| Retinol | • Increased serum RBP4 levels. | [39,43] |
| Insulin | • Increased risk of developing type II diabetes. | [38,44] |
| Adipokines | • Lower levels of adiponectin and resisting in hypothyroid patients compared to hyperthyroid patients. | [45] |

## 4. Discussion

The present study is the first to report adult-onset hypothyroidism in mice with deletion of the *Crabp1* gene, introducing a surprising possible role for *Crabp1* gene function in impacting thyroid state. The decrease in body temperature and increase in body weight beginning in adulthood around 3–4 months old support the adult-onset timing of hypothyroidism. Moreover, the plasma thyroid function test results (i.e., increased TSH and decreased T4 and T3 levels) are consistent with the pattern seen in primary hypothyroidism. The histopathology and the gene expression studies show alterations that may shed light on the pathogenesis of CKO hypothyroidism; alternatively, some of these changes might be secondary to the hypothyroid state. Additionally, understanding the high reverse T3, which is not usually seen in uncomplicated primary hypothyroidism, may shed light on the role of CRABP1. The possible mechanism of atRA to maintain thyrocyte function through CRABP1-mediated noncanonical activity would need further study. The resemblance of the CKO hypothyroidism phenotype to human adult hypothyroidism suggests that CKO mice can serve as an animal model for studying human adult primary hypothyroidism.

### 4.1. Physiological and Hormonal Changes in Animal Models of Primary Hypothyroidism or Altered Thyroid Metabolism

The physiological changes in CKO male mice are not entirely identical to other mouse models of hypothyroidism. Hypothyroid CKO mice have significantly smaller thyroid glands than their W.T. counterparts. This has been described in congenitally hypothyroid *hyt⁻/hyt⁻* (TSH-R mutation) mice but not in other models of primary hypothyroidism [46]. In contrast, TPO mutant [47], thyroglobulin mutant (*cog⁻/cog⁻*) [48], NIS [49], DUOX [50], and SLC26A7 [51] knockout mice have enlarged thyroid glands.

Other mouse models of hypothyroidism are typically associated with dwarfism or decreased growth [46–48,50] or no difference in body weight [49,52], in contrast to the adult onset in body weight seen in CKO male mice.

In the MCT-8 deletion model of peripheral tissue-specific hypothyroidism, no difference was detected in body weight, but these models developed a goitrous thyroid gland at 4 months old [52]. Their plasma thyroid hormone levels do not show the pattern of primary hypothyroidism.

Therefore, while deleting certain genes involved in thyroid hormone homeostasis could cause a phenotype to partially mimic hypothyroidism, they did not recapitulate adult-onset human hypothyroidism. As such, CKO mice may be more appropriate for modelling human adult-onset primary hypothyroidism. The timeline of the observed temperature change, weight gain, and hormonal changes in CKO mice, i.e., elevated TSH and lowered T3 and T4 plasma levels, confirm adult-onset primary hypothyroidism. Onset specific to adult age has not been reported in other mouse models of hypothyroidism.

### 4.2. Altered Gene Expression in CKO Mice Recapitulating Genetic Defects in Human Hypothyroidism

It may be relevant that most of the genes that were abnormally expressed in CKO mice have also been implicated in hypothyroid human patients (Table 3), suggesting that molecular defects caused by deleting *Crabp1* in mice are likely mimicking specific genetic

alterations contributing to the pathological development or disease progression of human adult hypothyroidism.

The interaction between thyroid state and other hormonal systems, in particular vitamin A (Table 4), is interesting. CRABP1 is the mediator of a wide spectrum of noncanonical activities of atRA, the principal active metabolite of vitamin A. Mouse *Crabp1* gene expression is tightly controlled by thyroid hormones. How these two hormonal systems interact, communicate, and cross-regulate would be an important topic in future studies.

**Author Contributions:** Conceptualization, L.-N.W. and F.N.; methodology, L.-N.W., F.N., C.-W.W. and L.M.; software, F.N.; validation, F.N., C.-W.W. and L.M.; formal analysis, F.N. and C.-W.W.; resources, L.-N.W. and D.S.; data curation, F.N., J.N., C.-W.W. and L.M.; writing—original draft preparation, L.-N.W., F.N. and J.N.; writing—review and editing, L.-N.W., F.N., J.N., C.-W.W. and L.M.; supervision, L.-N.W., L.B. and D.S. All authors have read and agreed to the published version of the manuscript.

**Funding:** J.N. is supported by 1F31DK123999. These studies were supported by NIH grants DK54733, DK60521, and the Dean's Commitment and the Distinguished McKnight Professorship of the University of Minnesota to LN.W.

**Institutional Review Board Statement:** All the studies have been approved by University of Minnesota IACUC (2203-39874A, approved 6 February 2022) and IBC (2012-38742H, approved 2 February 2022).

**Informed Consent Statement:** Not applicable.

**Data Availability Statement:** All data are available upon reasonable request to the corresponding author.

**Conflicts of Interest:** The authors declare no conflict of interest.

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
