# Peer review of "Deleting Cellular Retinoic-Acid-Binding Protein-1 (Crabp1) Gene Causes Adult-Onset Primary Hypothyroidism in Mice"

_endocrines, doi:10.3390/endocrines4010013_

Round 1
Reviewer 1 Report
The manuscript from Najjar et al reports very interesting findings from studies exploring the role of cellular retinoic acid binding protein-1 (Crabp-1) in maintaining thyroid hormone metabolism and actions. The authors have undertaken convincing experimental studies in a recently generated knockout mouse for Crabp1 and report human data from in silico investigations of hypothyroidism as it relates to retinoic acid metabolism and actions. The authors’ findings are very significant for understanding both thyroid hormone and retinoid actions in the body.
The manuscript is very well written and very informative. The experimental data are convincing that the absence Crabp1 expression adversely affects thyroid hormone metabolism and actions, giving rise to age-dependent hypothyroidism. These findings raise new and novel questions regarding how the thyroid hormone and retinoic acid hormone systems interact, communicate and cross-regulate each other.
Author Response
2023-2-12
Thanks to reviewer 1’s positive comments.
Reviewer 2 Report
The current original study emphasizes for the first time the crucial role of CRABP1 in the maintenance of the healthy adult thyroid gland and reports that CKO mice could be a reliable experimental animal model for studying the mechanisms involved in the development of adult hypothyroidism in humans. The Introduction, Results and Discussion parts are well-written and conclusive, with minor errors to be corrected, however the Materials and Methods part, respectively the Figure Legends need to be re-written in a more understandable manner as it is very difficult to follow if you are not very aware of those techniques.
I suggest English language quality improvement, maybe you could give it to a native speaker for the revision of language errors. Especially the Figure legends sections need to be revised thoroughly. I am enumerating some of the errors from the text, which are very eye striking, nevertheless there are others, too, if one is very fault-finder.
Line 34…as well as other what??
Line 37.. to THE active form
Line 78…all experimental procedures, no capital E is needed
Line 80 AT temperature instead of in temperature
Line 82 FROM the rectum instead of from a rectal
Line 88 the thyroid gland and surrounding trachea WERE harvested, study mice is not needed here
Lines 88-89 I would say: The major diameter and minor diameter for each lobe of the thyroid gland were measured,…
Line 90-91 I would say: The sum of THE two lobes was divided by THE body weight IN ORDER TO normalize the thyroid gland.
Lines 91-92 AFTERWARDS the lobes were isolated, instead of Followed by lobes were isolated.
Line 93 The RESULTED instead of resultant
Line 95. …and centrifuged AT 10000 g
Line 97 …according to THE FOLLOWING ELISA PROTOCOLS: (….
Line 101 Harvested thyroid GLANDS…
Line 103 HISTOPATHOLOGICAL evaluation
Line 106 reverse transcription kit …applied BIOSYSTEMS, THERMO Fisher Scientific
Line 107 Real time PCR reaction WAS PERFORMED in triplicates IN CASE of each sample.
Lines 108-109 The values were normalized to GAPDH. All primers are described in Table 4.
Line 133 plasma LEVELS
Line 140 I would leave out the word NOTION from this sentence. Either you write THESE DATA SUPPORT THAT or THESE DATA SUPPORT THE CONCEPT that.
Line 179 because OF their roles
Line 182 appeared NOT TO BE affected
Line 222 T-test WAS performed to…
Line 345-346…have enlarged thyroid glands a 2 month old hypothyroid??? What do you mean here?
. It not understandable.
Line 385 It is also INTERESTING that…
Overall, I believe that the manuscript delivers novel data, the methodology used is well-chosen and the experiments are well-planned, the results are conclusive and understandable. I suggest to accept the manuscript upon extensive editing of English language and style, especially of the sections mentioned above.
Author Response
Response to reviewer 2 (endocrines-2219291). 2023-2-12
- In response to reviewer 2, this revision has been extensively modified and edited according to the list of 2nd reviewer’s specific comments particularly about text sections from original line 34 to line 385.
Further, “Discussion” was extensively modified (see following point 2) to discuss published studies in comparison to our current study which is more specific to primary hypothyroidism. This will clarify and enhance the novelty of our animal model reported in this paper.
- We made several modifications to improve the clarity of our discussion/conclusion by deleting references that are not particularly relevant to primary hypothyroidism in adults, and by providing additional references that are more relevant for a comparison to our animal model that focuses on primary hypothyroidism in adults. These modifications will enhance the novelty and rigor of this study, and to strengthen our conclusion that this current study is the first to report a new animal model highly specific to “primary hypothyroidism in adults”.
These modifications do not alter the conclusion of this study. Added and deleted references related to these modifications are listed in the following.
- Newly added references. () indicates the # in the revised reference list.
Beamer WJ, Eicher EM, Maltais LJ, Southard JL. Science 212: 61–63, 1981. doi: 10.1126/science.7209519. (46)
Takabayashi S, Umeki K, Yamamoto E, Suzuki T, Okayama A, Katoh H. Mol Endocrinol 20: 2584–2590, 2006. doi: 10.1210/me.2006-0099. (47)
Beamer WG, Maltais LJ, DeBaets MH, Eicher EM. Endocrinology 120: 838–840, 1987. doi: 10.1210/endo-120-2-838. (48)
Grasberger H, De Deken X, Mayo OB, Raad H, Weiss M, Liao X-H, Refetoff S. Mol Endocrinol 26: 481–492, 2012. doi: 10.1210/me.2011-1320. (50)
Cangul H, Liao X-H, Schoenmakers E, Kero J, Barone S, Srichomkwun P, Iwayama H, Serra EG, Saglam H, Eren E, Tarim O, Nicholas AK, Zvetkova I, Anderson CA, Frankl FEK, Boelaert K, Ojaniemi M, Jääskeläinen J, Patyra K, Löf C, Williams ED, UK10K Consortium, Soleimani M, Barrett T, Maher ER, Chatterjee VK, Refetoff S, Schoenmakers N. JCI Insight 3, 2018. doi: 10.1172/jci.insight.99631. (51)
- Deleted references. () indicates the # in the original reference list.
Bianco AC, Kim BW. J Clin Invest. 2006;116: 2571–2579. doi:10.1172/JCI29812 (9)
Gaitonde DY, Rowley KD, Sweeney LB. Am Fam Physician. 2012;86: 244–251. doi:10.1080/20786204.2012.10874256 (10)
Chiovato L, Magri F, Carlé A. Adv Ther. 2019;36: 47–58. doi:10.1007/s12325-019-01080-8 (11)
Canani LH, Capp C, Dora JM, Meyer ELS, Wagner MS, Harney JW, Larsen PR, Gross JL, Bianco AC, Maia AL. J Clin Endocrinol Metab 90: 3472–3478, 2005. (39)
van der Deure WM, Hansen PS, Peeters RP, Uitterlinden AG, Fenger M, Kyvik KO, Hegedüs L, Visser TJ. Clin Endocrinol 70: 954–960, 2009. (44)
Peeters RP, van Toor H, Klootwijk W, de Rijke YB, Kuiper GGJM, Uitterlinden AG, Visser TJ. J Clin Endocrinol Metab 88: 2880–2888, 2003. (32)
Peeters RP, van der Deure WM, Visser TJ. Eur J Endocrinol 155: 655–662, 2006. (41)
Mentuccia D, Proietti-Pannunzi L, Tanner K, Bacci V, Pollin TI, Poehlman ET, Shuldiner AR, Celi FS. 2002. Doi:10.2337/diabetes.51.3.880
Schneider MJ, Fiering SN, Thai B, Wu S-Y, St Germain E, Parlow AF, St Germain DL, Galton VA. Endocrinology 147: 580–589, 2006. (57)
Schneider MJ, Fiering SN, Pallud SE, Parlow AF, St Germain DL, Galton VA. Mol Endocrinol 15: 2137–2148, 2001. (58)
Göthe S, Wang Z, Ng L, Kindblom JM, Barros AC, Ohlsson C, Vennström B, Forrest D. Genes Dev 13: 1329–1341, 1999. (59)
Christoffolete MA, Linardi CCG, de Jesus L, Ebina KN, Carvalho SD, Ribeiro MO, Rabelo R, Curcio C, Martins L, Kimura ET, Bianco AC. Diabetes 53: 577–584, 2004. (60)
Dumitrescu AM, Liao X-H, Weiss RE, Millen K, Refetoff S. Endocrinology 147: 4036–4043, 2006. (61)
Arthur JR, Beckett GJ, Mitchell JH. Nutr Res Rev 12: 55–73, 1999. (67)
Zimmermann MB, Jooste PL, Mabapa NS, Schoeman S, Biebinger R, Mushaphi LF, Mbhenyane X. Am J Clin Nutr 86: 1040–1044, 2007. (68)
Zimmermann MB, Wegmüller R, Zeder C, Chaouki N, Torresani T. J Clin Endocrinol Metab 89: 5441–5447, 2004. (69)
Rabbani E, Golgiri F, Janani L, Moradi N, Fallah S, Abiri B, Vafa M. Biol Trace Elem Res 199: 4074–4083, 2021. (70)